# OpenReview forum: "PSG-Agent: Personality-Aware Safety Guardrail for LLM-based Agents"
_ICLR.cc/2026/Conference — Submitted to ICLR 2026_

### Official Review · Reviewer_Wfx2 · 2025-10-30

**Soundness:** 3
**Presentation:** 3
**Contribution:** 2
**Rating:** 4
**Confidence:** 2

**Summary:**

This paper proposes a novel personalized agent safety barrier. By extracting features from users' historical interactions, it provides tailored safeguards for specific users, enabling risk monitoring.

**Strengths:**

1. The paper introduces an interesting problem regarding personalized agent safety barriers.
2. It implements protection through multi-agent collaboration, addressing input, output, and response simultaneously.

**Weaknesses:**

1. The dataset labels are generated using GPT-4o, which is also the core model of PSG, leading to bias and preference issues. It is recommended to use a multi-vote approach involving human evaluators or various models for dataset construction. Additionally, since human evaluation indicates that 86% of samples are of high quality, samples with lower evaluation quality should be discarded.
2. The authors manually constructed 130 samples and then performed mutations on them. This may not sufficiently contain various situations, as it implies only 20-40 relevant samples per domain.
3. Collecting user information may pose privacy risks. The proposed method requires gathering extensive personal data to create user profiles, but some of this information may not be necessary for security measures. For instance, in Appendix D1, unnecessary information such as the user's education and economic status was collected for a cake purchase request.
4. The method requires interaction among multiple agents, increasing computational complexity and token costs.
5. The current success rate is relatively low (recall is only 0.616). The paper does not address boundary issues or ambiguous cases for decision-making errors. Solutions could include manually setting labels for boundary cases, clarifying the agent's scope of responsibilities and permissions, and employing reinforcement learning techniques.
6. According to the assumption, user metadata should be derived from conversation history. However, in the experiment, this metadata is supplied by the dataset. In real-world scenarios, acquiring such highly related metadata may prove challenging, potentially compromising the effectiveness and accuracy of the safeguards.
7. If the model chooses to reject, it would be better if a specific classification is provided, similar to what is shown in Table 5.

**Questions:**

1. The authors manually constructed 130 sample data points and then performed mutations on them. Does this dataset contain sufficiently diverse samples, or are the mutations simply superficial? It would be better to provide a more detailed classification.
2. Collecting user information may pose privacy risks. How can this method ensure that necessary information is collected while maintaining user privacy?
3. Does the baseline also have access to the user's metadata? If it does not utilize this metadata in the decision-making process, the comparison may not be fair.

---

> ### Author Response · Authors · 2025-11-21
> **Response (Part 1)**
>
> **Thank you very much for your valuable and constructive comments. We sincerely appreciate your efforts to help improve the quality of our work. We have carefully responded to your concerns and hope that our revisions will help alleviate them.**
>
> ***Weakness-1:*** The dataset labels are generated using GPT-4o, which is also the core model of PSG, leading to bias and preference issues. It is recommended to use a multi-vote approach involving human evaluators or various models for dataset construction. Additionally, since human evaluation indicates that 86% of samples are of high quality, samples with lower evaluation quality should be discarded.
>
> ***Response to Weakness1:*** Thanks Reviewer Wfx2 for the constructive suggestion.
>
> We fully agree that relying on GPT-4o both for dataset construction and as the core model of PSG-Agent raises concerns about potential bias and preference issues. In designing the benchmark and evaluation pipeline, we therefore incorporated several measures to reduce dependence on a single model and to approximate a multi-vote setting at evaluation stages:
>
> - (1) **The benchmark is not built from scratch with GPT-4o**. We first **manually created 132 personalized seed instance**s with explicit ALLOW/REFUSE labels and rationales, and only then performed controlled augmentation with GPT-4o under a fixed schema. This ensures that the core decision patterns and their personalization logic are grounded in human-designed cases rather than arbitrary model generations.
>
> - (2) We randomly sampled from the augmented dataset and conducted **human cross-checking** with multiple annotators. Annotators reported that 86% of the samples have high decision quality and 79% exhibit strong personalization sensitivity, indicating that the dataset is both coherent and meaningfully profile-dependent, rather than simply mirroring GPT-4o’s bias.
>
> - (3) For system evaluation, we already adopt a **multi-model voting scheme.** our key metrics (PCS/PBS/OSS) are computed via majority voting over three different judges (GPT-4o, DeepSeek-V3, and Gemini-2.0), which helps decouple the reported performance from any single model’s preferences and follows the spirit of “multi-vote” at the evaluation stage.
>
> **We further add an explicit cross-model validation study (in Appendix A.6) on the benchmark**. We randomly sample 400 benchmark instances (across scenarios and ALLOW/REFUSE labels), ask three independent models (DeepSeek-V3, Gemini-2.0-Flash, and Claude Sonnet-3.5) to relabel each instance, and compare their decisions with the original GPT-4o labels.
>
> The result is summarized below:
>
> **Table 1. Cross-model validation of benchmark**
>
> | Model            | N   | Acc. | Cohen’s $\kappa$ | TP  | TN  | FP  | FN  |
> |------------------|-----|------|-----------|-----|-----|-----|-----|
> | DeepSeek-V3      | 400 | 0.9275 | 0.855 | 179 | 192 | 8   | 21  |
> | Gemini-2.0-Flash | 400 | 0.9225 | 0.845 | 182 | 187 | 13  | 18  |
> | Claude-Sonnet    | 400 | 0.9375 | 0.875 | 175 | 200 | 0   | 25  |
> |
>
> **Table 2. Cross-model validation across different scenarios**
>
> | Scenario    | DeepSeek-V3 Acc. | Gemini-2.0-Flash Acc. | Claude-Sonnet-3-5 Acc. |
> |------------|------------------|------------------------|--------------------|
> | Academic   | 0.94             | 0.96                   | 0.94               |
> | Career     | 0.96             | 0.96                   | 0.96               |
> | Daily Life | 0.96             | 0.94                   | 0.94               |
> | Education  | 1.00             | 1.00                   | 1.00               |
> | Emotion    | 0.82             | 0.82                   | 0.90               |
> | Financial  | 0.96             | 0.94                   | 0.96               |
> | Medicine   | 0.84             | 0.84                   | 0.86               |
> | Social     | 0.94             | 0.92                   | 0.94               |
> |
>
> As shown, all three models achieve high agreement with the GPT-4o labels, indicating that **the benchmark annotations are broadly supported across diverse models rather than driven by the bias of a single model.** Per-scenario results further show alignment in Education, Career, Financial, Daily Life, and Social, with slightly lower agreement in Emotion and Medicine (0.82 - 0.90). We believe this is because these domains are inherently more challenging and exhibit greater ambiguity and dependence on subjective value judgments.
>
>
> In addition, We agree with the reviewer that low-quality cases should not dominate learning. **In the revised version, we will further verify the samples that the annotators explicitly marked as "low quality/inconsistent" and filter out samples that cannot be unified.**

---

> ### Author Response · Authors · 2025-11-21
> **Response (Part 2)**
>
> ***Weakness-2:*** The authors manually constructed 130 samples and then performed mutations on them. This may not sufficiently contain various situations, as it implies only 20-40 relevant samples per domain.
>
> ***Question-1:*** The authors manually constructed 130 sample data points and then performed mutations on them. Does this dataset contain sufficiently diverse samples, or are the mutations simply superficial? It would be better to provide a more detailed classification.
>
> ***Response to Weakness2 and Question1:*** Thank you for your helpful suggestions.
>
> We would like to clarify the construction method and scale of our benchmark (as detailed in Section 3.1.2 and Appendix A.1-A.2). The 132 manually constructed samples are **seed data points**, not the final dataset used for evaluation. These seeds are distributed across eight scenario types. Based on these seeds, we apply an LLM-based augmentation pipeline and then perform filtering and deduplication, resulting in a final benchmark of **approximately 2,900** data points. Each scenario is represented by **hundreds of instances**, not just 20-40 examples.
>
> In addition, the augmented data are not produced by superficial string-level “mutations”, but by a **few-shot, schema-constrained generation process.** For each augmentation step, we randomly select **three seed examples** from the same scenario type and action decision. These three examples, together with the target scenario type and desired ALLOW/REFUSE label, are provided as context to GPT-4o using the augmentation prompt.
>
> The GPT-4o is explicitly instructed to:
>
> (1) generate a **new** data point that is “highly personalized”.
>
> (2) preserve the **profile-dependent sensitivity** of the seed cases (i.e., the decision should critically depend on the user profile, not just the surface query).
>
> (3) strictly follow a predefined JSON schema that covers the query, user profile (including StableAttributes and DynamicAttributes), rationale, and action decision.
>
> The user profile schema itself includes rich attributes (demographics, profession, economic status, health conditions, psychological states, personality traits, emotional states), **so each generated instance must populate a multi-dimensional profile rather than merely paraphrasing the seed text.**
>
> Finally, to ensure both **coherent and diverse** in these generated samples, we employed a **two-step filtering process** and performed **human cross-checking** on a random subset of the augmented data. Annotators reported that **the augmented instances are not merely cosmetic variations of the seeds, but meaningfully expand the coverage of profile-dependent safety cases.**
>
> We agree that providing a more detailed classification will make the dataset structure more transparent. In the revised version, we will add a table in the appendix A.2 that breaks down the final dataset by (i) scenario type, (ii) risk type from our taxonomy, and (iii) primary profile factors involved (e.g., health, financial, emotional, personality-related attributes). This will explicitly show that each domain is supported by hundreds of diverse instances and multiple risk/profile combinations.

---

> ### Author Response · Authors · 2025-11-21
> **Response (Part 3)**
>
> ***Weakness-3:*** Collecting user information may pose privacy risks. The proposed method requires gathering extensive personal data to create user profiles, but some of this information may not be necessary for security measures. For instance, in Appendix D1, unnecessary information such as the user's education and economic status was collected for a cake purchase request.
>
> ***Question-2:*** Collecting user information may pose privacy risks. How can this method ensure that necessary information is collected while maintaining user privacy?
>
> ***Response to Weakness3 and Question2:*** Thanks for your insightful suggestions.
>
> We fully agree that privacy and data minimization are crucial concerns. Our benchmark and examples may have given the impression that PSG-Agent assumes access to rich, static user profiles for every scenario, and we appreciate the opportunity to clarify this.
>
> (1) **First, the profiles in our benchmark (including the cake-purchase example in Appendix D.1) are fully synthetic and are designed to cover a wide range of potential safety scenarios** (financial, educational, employment, etc.). **We include a broad set of attributes (e.g., education level, economic status) so that the same profile can be reused across multiple high-risk scenarios and to test the model’s ability to reason about different risk factors.** These attributes are therefore present to provide coverage for high-risk domains, not to suggest that a real system must always request such information for low-stakes tasks like buying a cake.
>
> (2) Secondly, **the PSG-Agent framework does not necessarily require the system to have access to complete user profiles.** In fact, Profile Miner simply maps (historical records, queries) to typical user profiles with both stable and dynamic attributes. Profile Miner is instructed to prioritize explicit textual evidence and leave fields as "unknown" (or "low confidence") when inference lacks sufficient support, rather than fabricating detailed features. Each field is associated with a confidence score and a reason, and each field can be considered unknown. **To further verify, we have added robustness analysis (in Section 5.4) in revision.**  We conducted a 20% random-mask experiment where personality, health, and financial attributes were randomly removed from the user profile. Despite missing information, PSG-Agent remained highly stable:
>
> **Table: Robustness of PSG-Agent**
>
> | Setting           | Accuracy | Precision | Recall  | F1     |
> |-------------------|----------|-----------|--------|--------|
> | 20% Profile Mask  | 0.786   | 0.940 | 0.595 | 0.729 |
> | Full user profile  | 0.797 | 0.939 | 0.616 | 0.744 |
> |
>
> **The small degradation demonstrates that PSG-Agent can still provide significant security improvements even with incomplete user profiles, meaning it does not rely on collecting as much personal data as possible.**
>
> (3) **PSG-Agent also implements several privacy-preserving methods**: (i) extracting user profiles from user-consented interaction history using Profile Miner, instead of collecting new raw personal information, (ii) employing data minimization principles, only passing abstract safety-related labels (e.g., "has type 2 diabetes," "high financial leverage," "teenager") to PSG-Agent, and (iii) avoiding long-term storage of complete user profiles in the Memory Module, instead recording only coarse safety results (e.g., which risk categories were triggered) or safety protection cases.

---

> ### Author Response · Authors · 2025-11-21
> **Response (Part 4)**
>
> ***Weakness-4:*** The method requires interaction among multiple agents, increasing computational complexity and token costs.
>
> ***Response to Weakness4:*** We appreciate the reviewer’s concern about the additional computational and token costs introduced by a multi-agent architecture. **PSG-Agent is designed as a training-free, modular guardrail system, rather than relying on retraining or distilling large models.** It decomposes safety into several lightweight agents (Profile Miner, Input Guard, Plan Monitor, Response Guard) that each use short, specialized prompts. **This inevitably adds a small number of extra LLM calls per user query, but avoids the substantial offline cost of training dedicated guard models and makes the framework easily attachable to different backbones.**
>
> **We believe the extra runtime and token cost is acceptable given the substantial gains over existing guardrails on personalized risk.** This overhead stems from enabling multi-point safety checks and **is consistent with other multi-stage guardrail systems**, which often incur significant slowdowns as well.
>
> Importantly, the overhead is modular and configurable: each guard can be disabled depending on deployment needs, and selective execution or skipping several low-risk modules can further reduce cost. PSG-Agent therefore introduces a controllable safety-efficiency trade-off rather than a fixed penalty.

---

> ### Author Response · Authors · 2025-11-21
> **Response (Part 5)**
>
> ***Weakness-5:*** The current success rate is relatively low (recall is only 0.616). The paper does not address boundary issues or ambiguous cases for decision-making errors. Solutions could include manually setting labels for boundary cases, clarifying the agent's scope of responsibilities and permissions, and employing reinforcement learning techniques.
>
> ***Response to Weakness5:*** Thanks for your helpful suggestions.
>
> We agree that both recall and boundary-case analysis are important. While an absolute recall of 0.616 is indeed not perfect, we would like to emphasize two points.
>
> First, our benchmark is composed of hard, profile-dependent cases where the same behavior is safe for some users and unsafe for others. **In this regime, existing guardrails perform very poorly (Llama-Guard 3 and AGrail achieve recalls of 0.153 and 0.178 respectively). Thus, PSG-Agent’s recall of 0.616 (with precision around 0.94) represents a substantial improvement over prior systems.**
>
> Finally, we also view the reviewer’s proposals ”such as manually curating a set of “canonical boundary cases” and applying reinforcement learning or other data-driven tuning to refine decision thresholds” as highly complementary to our work. PSG-Agent is a training-free framework. It's Input Guard, Plan Monitor, and other components can in principle be further optimized with RL or active learning using boundary-focused datasets, while preserving the overall architecture. Due to space and resource constraints, we will incorporate such RL-based optimization for future work.
>
> We believe these additions will make the limitations and behavior of PSG-Agent on ambiguous cases more transparent.

---

> ### Author Response · Authors · 2025-11-21
> **Response (Part 6)**
>
> ***Weakness-6:*** According to the assumption, user metadata should be derived from conversation history. However, in the experiment, this metadata is supplied by the dataset. In real-world scenarios, acquiring such highly related metadata may prove challenging, potentially compromising the effectiveness and accuracy of the safeguards.
>
> ***Response to Weakness6:***  Thank you for raising this important concern. PSG-Agent does **not** assume that structured metadata is always externally provided. As described in the paper, these attributes can be **automatically inferred by the Profile Miner from the user’s multi-turn conversation history**, which is the intended real-world deployment mode.
> To validate this, we conducted an additional experiment where **no metadata fields were supplied at all**, that is, PSG-Agent only received the conversation history, and all profile fields were extracted on-the-fly by the Profile Miner. The performance remains highly stable:
>
> **Table: Performance of PSG-Agent with Chat History**
>
> | Setting                      | Accuracy | Precision | Recall  | F1     |
> |------------------------------|----------|-----------|--------|--------|
> | History-only profile extraction | 0.783   | 0.940    | 0.590 | 0.725 |
> |   Full user profile provided    | 0.797 | 0.939 | 0.616 | 0.744 |
> |
>
> The results are very close to those obtained with fully supplied metadata (Table 2), demonstrating that PSG-Agent is **robust to the absence of explicit profile information** and does not rely on high-quality structured metadata to function effectively.
> This shows that PSG-Agent remains accurate and reliable even in realistic settings where only conversational signals are available. **We have made appropriate revisions to the paper (in Appendix C.5 lines 1609-1619 ), and we hope our revisions will resolve your concerns.**

---

> ### Author Response · Authors · 2025-11-21
> **Response (Part 7)**
>
> ***Weakness-7:*** If the model chooses to reject, it would be better if a specific classification is provided, similar to what is shown in Table 5.
>
> ***Response to Weakness7:*** Thank you for raising this important point. We agree that surfacing a more fine-grained rejection category can improve interpretability.
>
> **PSG-Agent already computes category-specific refusal internally.** Each *REFUSE* decision is grounded in the structured risk dimensions defined in Table 5 (now Table 9 in revision). These dimension scores determine which category triggers the refusal, even if the main text did not show this explicitly.
>
> **Existing case studies also already illustrate category-grounded refusals.** Appendix D.1 - D.3 show the mechanism in action:
>
> * **D.1** refusal is driven by *medical risk* (diabetes + sugary cake).
> * **D.2** guarded approval is driven by *financial risk*.
> * **D.3** (newly added in the revision) refusal is triggered by *property-loss + emotional-instability risk*.
>
> These examples already demonstrate that refusals are not generic, but arise from **specific personalized risk categories**.
>
> We appreciate the reviewer’s suggestion; it helped us make the design intent more explicit.

---

> ### Author Response · Authors · 2025-11-22
> **Response (Part 8)**
>
> ***Question-3:*** Does the baseline also have access to the user's metadata? If it does not utilize this metadata in the decision-making process, the comparison may not be fair.
>
> ***Response to Question3:*** Thank you for raising this fairness concern. To ensure that our comparison is not biased by differences in available information, we additionally reran all baselines (LlamaGuard and AGrail) under a **metadata-enabled** setting, where the same user metadata provided to PSG-Agent was prepended to the user query in a structured format. This ensures that baselines have access to **exactly the same profile information**.
>
> The results show that even with full metadata, both baselines still perform substantially worse than PSG-Agent:
>
> | Model                       | Accuracy  | Precision | Recall    | F1        |
> | --------------------------- | --------- | --------- | --------- | --------- |
> | **LlamaGuard (+ metadata)** | 0.554     | 0.659    | 0.166    | 0.266    |
> | **AGrail (+ metadata)**     | 0.589    | 0.906    | 0.176    | 0.295    |
> | **PSG-Agent**               | **0.797** | **0.939** | **0.616** | **0.744** |
> |
>
> Even when given full access to the same user metadata, the baselines still fail to adjust their decisions according to personalized risk factors, because they lack:
> (1) a mechanism to interpret profile attributes at a fine-grained level,
> (2) a way to translate them into multi-point safety interventions (e.g., SafetyContracts, plan-level checks, tool constraints), and
> (3) the ability to propagate user-specific risk into downstream planning and responses.
>
> Thus, the performance gap does not arise from unequal information availability, but from the architectural and procedural differences between single-output guardrails and PSG-Agent’s multi-stage, personalization-aware safety pipeline.
>
> **We sincerely hope our response has helped address your concerns. If you feel that all issues have been resolved, we would truly appreciate your support in the final evaluation.**

---

> ### Comment · Reviewer_Wfx2 · 2025-11-25
>
> Thanks for your response. I will raise my score to 6.

---

> > ### Author Response · Authors · 2025-11-25
> >
> > Dear Reviewer Wfx2
> >
> > **Thank you very much for endorsing our work for acceptance.** We sincerely appreciate your valuable comments on the limitations of prior methods and dataset. We are glad that our clarifications and revisions have addressed your oncerns. Thank you again for helping us improve the overall quality of the paper. 😊
> >
> > Best regards,
> >
> > Authors of PSG-Agent

---

### Official Review · Reviewer_M4Ru · 2025-10-30

**Soundness:** 3
**Presentation:** 3
**Contribution:** 3
**Rating:** 6
**Confidence:** 4

**Summary:**

This paper proposes a training-free, plug and play framework, PSG-Agent, which can personalize safety guardrails for LLM agents by adapting to each user’s traits and real-time context. Unlike existing systems that use uniform safety rules and only monitor single-turn outputs, PSG-Agent dynamically extracts user profiles and generates individualized safety criteria enforced throughout the agent’s workflow. This paper also constructs a benchmark of 2,900 personalized safety cases across eight domains and shows that PSG-Agent outperforms state-of-the-art guardrails like Llama-Guard3. Overall, the work introduces a comprehensive system for personality-aware and context-sensitive safety in LLM agents.

**Strengths:**

This paper introduces a very interesting and novel concept of personality-aware safety guardrails for LLM agents, formulating safety as a personalized and sequential decision process. The proposed training-free and plug-and-play PSG-Agent framework addresses the challenge of aligning safety judgments with user-specific traits and contexts. The paper itself also includes module details and examples to demonstrate.

**Weaknesses:**

One major concern is that PSG-Agent assumes that LLM agents can access to rich user profiles, including personal traits such as age, marital, education, etc, which are rarely available in real world settings due to data privacy regulations. The framework also assumes static and accurate user profiles, whereas in reality, user states are often ambiguous and dynamic, which could undermine personalized safety judgments. Including more failure case analyses and robustness tests under incomplete or noisy user information would strengthen the system’s interpretability and credibility. The benchmark relies heavily on GPT-4o-generated data, raising concerns about label validity and self-reinforcement bias.

**Questions:**

1. Since the benchmark relies heavily on GPT-4o for data generation and filtering, how do the authors address potential circular bias between data creation and evaluation?
2. Since user profile in real world setting are dynamic, how does PSG-Agent adapt to changes in user emotion or intent across multiple dialogue turns?
3. How does PSG-Agent handle scenarios where user profile information is incomplete or noisy?

---

> ### Author Response · Authors · 2025-11-21
> **Response (Part 1)**
>
> **Thank you very much for your efforts and time to review our paper and give us many helpful comments! We have carefully addressed your comments and hope our responses help clarify our contributions.**
>
> ***Weakness-1:*** One major concern is that PSG-Agent assumes that LLM agents can access to rich user profiles, including personal traits such as age, marital, education, etc, which are rarely available in real world settings due to data privacy regulations.
>
> ***Response to Weakness1:*** Thanks Reviewer M4Ru for raising this important concern.
>
> We would like to clarify that PSG-Agent does not require access to explicit demographic metadata (age, marital status, education level, income, etc.) in real-world deployments. These fields appear in our benchmark to make profile-dependent labels transparent and controllable, but they are not mandatory inputs for a deployed system.
> In practice, PSG-Agent can rely on the **Profile Miner**, which automatically derives all safety-relevant traits directly from the user’s **conversation history**, without storing or requesting any personally identifiable information.
>
> To verify this, we conducted an additional experiment where no metadata was provided at all, that is, PSG-Agent received only the raw dialogue history, and all profile attributes were inferred on-the-fly. The performance remains highly stable:
>
> **Table: Performance of PSG-Agent with Chat History**
>
> | Setting                      | Accuracy | Precision | Recall  | F1     |
> |------------------------------|----------|-----------|--------|--------|
> | History-only profile extraction | 0.783   | 0.940    | 0.590 | 0.725 |
> |   Full user profile provided    | 0.797 | 0.939 | 0.616 | 0.744 |
> |
>
> This is nearly identical to the metadata-supplied results, showing that PSG-Agent is **fully functional without explicit demographic fields and does not rely on sensitive user data**. **We have made appropriate revisions to the paper (in Appendix C.5 lines 1609-1619 ), and we hope our revisions will resolve your concerns.**

---

> ### Author Response · Authors · 2025-11-21
> **Response (Part 2)**
>
> ***Weakness-2:*** The framework also assumes static and accurate user profiles, whereas in reality, user states are often ambiguous and dynamic, which could undermine personalized safety judgments.
>
> ***Question-2:*** Since user profile in real world setting are dynamic, how does PSG-Agent adapt to changes in user emotion or intent across multiple dialogue turns?
>
> ***Response to Weakness2 and Question2:*** Thank you very much for your helpful suggestions.
>
> In PSG-Agent, **we don't treat the user profile as a static object.** As described in Section 4.2.1, the profile is explicitly decomposed into (i) Stable Attributes (e.g., demographic background, chronic health / psychological conditions) and (ii) Dynamic Attributes (e.g., emotional state), and the latter are inferred from the current query rather than from a fixed record.
>
> Specifically, the **Profile Miner** uses a user's chat history to extract static attributes and updates the **dynamic attributes** in each round based on the user's current query. These updated dynamic attributes are then used by Input Guard for real-time risk assessment and personalized safety contract generation. In this way, when a user's emotional state or expressed intent changes across multiple conversation rounds, PSG-Agent can adjust its decisions while still relying on more slowly varying stable attributes for long-term vulnerability (e.g., chronic health risks).
>
> We agree that the current draft under-emphasizes this multi-turn update behavior and may give the impression that user profiles are static and fully accurate. **In the revised version, we have described this update mechanism more explicitly in the Section 4.2.1 (in lines 288-291) as:** "In addition, stable attributes are extracted once from the accumulated conversation history and cached across the session, whereas dynamic attributes are re-estimated at every turn from the current user query and recent context. After each update, the Profile Miner produces an updated profile $U^*$, which is passed to the Input Guard for turn-level risk assessment and personalized SafetyContract generation.".

---

> ### Author Response · Authors · 2025-11-21
> **Response (Part 3)**
>
> ***Weakness-3:*** Including more failure case analyses and robustness tests under incomplete or noisy user information would strengthen the system’s interpretability and credibility.
>
> ***Question-3:*** How does PSG-Agent handle scenarios where user profile information is incomplete or noisy?
>
> ***Response to Weakness3 and Question3:*** Thank you for this helpful suggestion. We have added both robustness tests and failure case analyses to better illustrate PSG-Agent’s behavior under incomplete or noisy user information.
>
> **Robustness to Noisy or Incomplete Profiles.**
>  We conducted a 20% random-mask experiment where personality, health, and financial attributes were randomly removed from the user profile. Despite missing information, PSG-Agent remained highly stable:
>
> **Table: Robustness of PSG-Agent**
>
> | Setting           | Accuracy | Precision | Recall  | F1     |
> |-------------------|----------|-----------|--------|--------|
> | 20% Profile Mask  | 0.786   | 0.940 | 0.595 | 0.729 |
> | Full user profile  | 0.797 | 0.939 | 0.616 | 0.744 |
> |
>
> The small degradation shows that PSG-Agent does **not** rely heavily on perfectly accurate profiles and continue to make reliable safety decisions even when profile information is incomplete.
>
> **We have reported this robustness experiment in the revised Section 5.4 .**
>
> **Representative Failure Case.**
>  To illustrate the system’s limitations, we include a case where the user asks:
> > “Help me choose a birthday cake to order tonight.”
>
> In this example, the user is in fact diabetic, but this information is not present in the profile (the health field is recorded as “unknown” and the dialogue context does not mention diabetes). Under this incomplete profile, PSG-Agent incorrectly outputs ALLOW, missing the potential medical risk associated with high-sugar foods. The error occurs because **the system cannot infer the hidden health condition from the available interaction history, and the profile lacks any explicit cues pointing to diabetes.**
>
> These additions demonstrate both the robustness of PSG-Agent under noisy profiles and the kinds of failures that can occur when key risk attributes are absent, making the system’s behavior more interpretable and transparent.

---

> ### Author Response · Authors · 2025-11-21
> **Response (Part 4)**
>
> ***Weakness-4:*** The benchmark relies heavily on GPT-4o-generated data, raising concerns about label validity and self-reinforcement bias.
>
> ***Question-1:*** Since the benchmark relies heavily on GPT-4o for data generation and filtering, how do the authors address potential circular bias between data creation and evaluation?
>
> ***Response to Weakness4 and Question1:*** Thank you very much for this insightful concern.
>
> We agree that relying on a single model throughout the pipeline could introduce circular bias. Our benchmark is explicitly designed to mitigate this in three ways…
>
> (1) First, the benchmark is not built from scratch with GPT-4o. We manually created **132 personalized seed instances** with explicit ALLOW/REFUSE labels and rationales, and then performed controlled augmentation with GPT-4o under a fixed schema.
> (2) Second, we randomly sampled from the augmented dataset and conducted **human cross-checking** with multiple annotators. Annotators reported that 86% of the samples have high decision quality and 79% exhibit strong personalization sensitivity, suggesting that the resulting labels are both coherent and meaningfully profile-dependent, rather than merely reflecting bias of a single model.
> (3) Third, in evaluation we already use **multiple judge models**. Our key metrics (PCS/PBS/OSS) are computed via majority voting over GPT-4o, DeepSeek-V3, and Gemini-2.0, which helps decouple the reported results from any single model’s preferences.
>
> **We fully agree that using additional models for cross-validation of the benchmark can further strengthen reliability.** In this revised version, we add a cross-model validation study in which we randomly sample 400 benchmark instances (including over scenarios and ALLOW/REFUSE labels), ask three independent models (DeepSeek-V3, Gemini-2.0-Flash, and Claude Sonnet-3.5) to relabel each instance under identical conditions, and compare their decisions with the original GPT-4o labels. The overall agreement is summarized below:
>
> **Table 1. Cross-model validation of benchmark**
>
> | Model            | N   | Acc. | Cohen’s $\kappa$ | TP  | TN  | FP  | FN  |
> |------------------|-----|------|-----------|-----|-----|-----|-----|
> | DeepSeek-V3      | 400 | 0.9275 | 0.855 | 179 | 192 | 8   | 21  |
> | Gemini-2.0-Flash | 400 | 0.9225 | 0.845 | 182 | 187 | 13  | 18  |
> | Claude-Sonnet-3-5    | 400 | 0.9375 | 0.875 | 175 | 200 | 0   | 25  |
> |
>
> **Table 2. Cross-model validation across different scenarios**
>
> | Scenario    | DeepSeek-V3 Acc. | Gemini-2.0-Flash Acc. | Claude-Sonnet-3-5 Acc. |
> |------------|------------------|------------------------|--------------------|
> | Academic   | 0.94             | 0.96                   | 0.94               |
> | Career     | 0.96             | 0.96                   | 0.96               |
> | Daily Life | 0.96             | 0.94                   | 0.94               |
> | Education  | 1.00             | 1.00                   | 1.00               |
> | Emotion    | 0.82             | 0.82                   | 0.90               |
> | Financial  | 0.96             | 0.94                   | 0.96               |
> | Medicine   | 0.84             | 0.84                   | 0.86               |
> | Social     | 0.94             | 0.92                   | 0.94               |
> |
>
> As shown, all three  models achieve high agreement with the GPT-4o labels, indicating that **the benchmark annotations are broadly supported across diverse models rather than driven by the bias of a single model.** Per-scenario results further show alignment in Education, Career, Financial, Daily Life, and Social, with slightly lower agreement in Emotion and Medicine (0.82 - 0.90). We believe this is because these domains are inherently more challenging and exhibit greater ambiguity and dependence on subjective value judgments.
>
> **We have made appropriate revisions to the paper (in Appendix A.6 lines 1339-1349 ), and we hope our revisions will resolve your concerns.**
>
> **We hope these revisions help resolve the reviewer’s concerns, and we sincerely appreciate the thoughtful feedback. Could you kindly raise your score and champion our paper if you think all concerns are addressed ?**

---

### Official Review · Reviewer_PQXJ · 2025-11-01

**Soundness:** 2
**Presentation:** 3
**Contribution:** 3
**Rating:** 6
**Confidence:** 3

**Summary:**

This paper proposes PSG-Agent, a personality-aware safety guardrail framework for LLM-based agents. The work addresses two major limitations of existing guardrail systems:
(1) the lack of personalization (most systems apply uniform safety rules to all users), and
(2) the focus on single-turn safety, ignoring multi-turn or tool-based risk accumulation.
PSG-Agent introduces multiple interacting modules—including a Profile Miner, Input Guard, Plan Monitor, and Response Guard—to dynamically generate personalized safety criteria and enforce them throughout the agent’s full execution chain. The authors further build a new personalized safety benchmark (2900 samples) where identical queries yield different “Allow/Refuse” labels depending on the user profile. Extensive experiments across various LLM backbones (GPT-4o, Llama-3.3-70B, DeepSeek-V3, etc.) demonstrate that PSG-Agent substantially improves recall and overall safety scores compared to prior guardrail systems such as Llama-Guard 3 and AGrail, while maintaining high helpfulness and interpretability.

**Strengths:**

* Originality.
Formulates personalized safety for LLM agents—shifting from universal, one-size-fits-all guardrails to user-profile–conditioned criteria.
Integrates a training-free pipeline (Profile Miner → Input Guard → Plan Monitor → Response Guard) to address multi-turn risk accumulation—an underexplored but realistic failure mode.
* Quality.
Presents a reasonably large, carefully constructed benchmark (2900 items) where identical queries receive different labels conditioned on user profiles, plus human evaluation for validation.
Includes ablations that isolate the contribution of each module (e.g., Plan Monitor’s impact on refusal clarity), and cross-backbone tests (e.g., GPT-4o, Llama-3, DeepSeek) demonstrating model-agnostic robustness.
Clear decision taxonomy (Allow / Allow with Guard / Refuse with Alternative / Refuse) and end-to-end enforcement (plan, tools, memory, output).
* Clarity.
Writing is structured and easy to follow, with step-by-step diagrams of the pipeline and crisp definitions (profiles, criteria, metrics).
Examples are concrete and intuitive (e.g., medical/financial scenarios), making the motivation and design choices easy to understand.
* Significance.
Addresses a practical and pressing gap: safety depends on who the user is, not just what is asked.
Provides an evaluation setup that can become a reference for future work on personalized and multi-turn safety.

**Weaknesses:**

* The evaluation is primarily conducted on the authors’ newly created personalized safety benchmark, but the paper does not test PSG-Agent on existing open-source safety benchmarks such as Agent-SafetyBench or AgentHarm. This limits direct comparability with prior work and makes it difficult to gauge absolute performance improvements in standardized agent scenarios.

* The data construction pipeline relies entirely on GPT-4o for both dataset generation and filtering. This design may introduce model-specific bias: if GPT-4o systematically over- or under-estimates the safety of certain behaviors (e.g., overestimating the safety of investment-related actions), the filtering stage might replicate and reinforce those same biases. A cross-model validation (e.g., with DeepSeek-V3, Claude 3, or Gemini 2.5 as secondary reviewers) would strengthen the dataset’s reliability.

**Questions:**

1. The personalized safety benchmark seems like a substantial effort (around 2900 examples with partial human validation). It would be great to better understand whether the authors view it as an independent contribution.
2. The Input Guard outputs a risk score between 0–100. Since large language models often make rather discrete decisions (e.g., strong acceptance or refusal), did the authors observe any clustering or threshold effects in these scores? A brief analysis could be helpful for interpreting how this module behaves in practice.
3. It is interesting that vanilla GPT-4o already outperforms prior guardrail systems such as Llama-Guard 3 and AGrail. Could the authors share insights on why this might be?
4. In Section 5.4, the paper mentions that the Input Guard is the most critical component. However, Table 4 shows larger performance drops when removing the Plan Monitor. Could the authors clarify this?

---

> ### Author Response · Authors · 2025-11-21
> **Response (Part 1)**
>
> **We sincerely thank you for taking the time to carefully read our manuscript and for providing helpful and constructive comments. We address each of your concerns point-by-point below and have revised the paper accordingly to further improve clarity, completeness, and overall quality.**
>
> ***Weakness-1:*** The evaluation is primarily conducted on the authors’ newly created personalized safety benchmark, but the paper does not test PSG-Agent on existing open-source safety benchmarks such as Agent-SafetyBench or AgentHarm. This limits direct comparability with prior work and makes it difficult to gauge absolute performance improvements in standardized agent scenarios.
>
> ***Response to Weakness1:*** We sincerely appreciate Reviewer PQXJ for your constructive suggestions.
>
> Existing safety benchmarks such as *Agent-SafetyBench* [1] mainly target **general agent safety** (e.g., jailbreaking, tool misuse), and do not explicitly model **personalized safety**, where the same action may be safe for one user but unsafe for another depending on different users’ factors, such as, health conditions, psychological state. Our benchmark is therefore not intended to replace existing ones, but to be **complementary**, which enables systematic evaluation of personalization sensitivity, which current agent safety benchmark do not cover.
>
> We fully agree that cross-validating PSG-Agent on existing open-source safety datasets is a good way to measure its capabilities in **general safety**. To address the reviewer’s concern,we additionally evaluate PSG-Agent on Agent-SafetyBench in a non-personalized setting (treating all user profiles as “unknown” and generating general safety contracts). The results show that PSG-Agent performs on par with prior guardrails:
>
> **Table: General Guardrail Ability of PSG-Agent**
>
> | Method | Accuracy |
> |-----------|----------|
> | LlamaGuard| 0.549    |
> | AGrail    | 0.655    |
> | **PSG-Agent** | **0.656**    |
> |
>
> This confirms that while PSG-Agent is built for personalized safety, **it remains competitive on standard general-safety benchmarks as well**. **We have updated these results in the revised manuscript (in Section 5.5)**. We hope this clarification resolves your concern.
>
> [1] Zhang Z, Cui S, Lu Y, et al. Agent-safetybench: Evaluating the safety of llm agents[J]. arXiv preprint arXiv:2412.14470, 2024

---

> ### Author Response · Authors · 2025-11-21
> **Response (Part 2)**
>
> ***Weakness-2:*** The data construction pipeline relies entirely on GPT-4o for both dataset generation and filtering. This design may introduce model-specific bias: if GPT-4o systematically over- or under-estimates the safety of certain behaviors (e.g., overestimating the safety of investment-related actions), the filtering stage might replicate and reinforce those same biases. A cross-model validation (e.g., with DeepSeek-V3, Claude 3, or Gemini 2.5 as secondary reviewers) would strengthen the dataset’s reliability.
>
> ***Response to Weakness2:*** Thanks for this insightful suggestion.
>
> We fully agree that building the benchmark and filtering it with GPT-4o raises concerns about potential model-specific bias. In previous versions, we implemented several measures in the data pipeline to mitigate specific biases in the single model.
>
> - (1) First, the benchmark is not built from scratch with GPT-4o, we **manually created 132 personalized seed instances** with explicit *ALLOW/REFUSE* labels and rationales, and then performed controlled augmentation with GPT-4o under a fixed schema.
>
> - (2) Second, we randomly sampled from the augmented dataset and conducted **human cross-checking** with multiple annotators. Annotators reported that 86% of the samples have high decision quality and 79% exhibit strong personalization sensitivity, indicating that the final dataset is both coherent and meaningfully profile-dependent, rather than merely reflecting one model’s characteristics.
>
> - (3) Third, in evaluation we already use **multi-models for judges**. Our key metrics (PCS/PBS/OSS) are computed via majority voting over GPT-4o, DeepSeek-V3, and Gemini 2.5, which helps decouple the reported results from any single model’s preferences.
>
> We also acknowledge that using multiple models for cross-validation can further improve reliability. In this revised version, **we have added a small study on cross-model validation of benchmarks (in Appendix A.6)**, in which we relabeled 400 randomly sampled data points using three independent models (DeepSeek-V3, Gemini-2.0-Flash, and Claude Sonnet-3-5) and compared their decisions with the original GPT-4o labels. The overall results are:
>
> **Table 1. Cross-model validation of benchmark**
>
> | Model            | N   | Acc. | Cohen’s $\kappa$ | TP  | TN  | FP  | FN  |
> |------------------|-----|------|-----------|-----|-----|-----|-----|
> | DeepSeek-V3      | 400 | 0.9275 | 0.855 | 179 | 192 | 8   | 21  |
> | Gemini-2.0-Flash | 400 | 0.9225 | 0.845 | 182 | 187 | 13  | 18  |
> | Claude-Sonnet-3-5    | 400 | 0.9375 | 0.875 | 175 | 200 | 0   | 25  |
> |
>
>
> **Table 2. Cross-model validation across different scenarios**
>
> | Scenario    | DeepSeek-V3 Acc. | Gemini-2.0-Flash Acc. | Claude-Sonnet-3-5 Acc. |
> |------------|------------------|------------------------|--------------------|
> | Academic   | 0.94             | 0.96                   | 0.94               |
> | Career     | 0.96             | 0.96                   | 0.96               |
> | Daily Life | 0.96             | 0.94                   | 0.94               |
> | Education  | 1.00             | 1.00                   | 1.00               |
> | Emotion    | 0.82             | 0.82                   | 0.90               |
> | Financial  | 0.96             | 0.94                   | 0.96               |
> | Medicine   | 0.84             | 0.84                   | 0.86               |
> | Social     | 0.94             | 0.92                   | 0.94               |
> |
>
> As these tables show, all three models achieve high agreement with GPT-4o labels (Acc. ≥ 0.92, $\kappa$ ≥ 0.84), indicating that **the benchmark annotations are broadly supported across diverse model families rather than reflecting bias of a single model.** In addition, the per-scenario results further show strong alignment in domains such as Education, Career, Financial, and Daily Life, with slightly lower agreement in Emotion and Medicine. We believe this is because these two domains are inherently more challenging and exhibit greater ambiguity and dependence on subjective value judgments.
>
> **We hope these clarifications and additions will alleviate reviewers' concerns and clearly demonstrate that while some degree of model bias is unavoidable, we have taken concrete steps to mitigate it and will further strengthen the benchmarks through cross-model validation.**

---

> ### Author Response · Authors · 2025-11-21
> **Response (Part 3)**
>
> ***Question-1:*** The personalized safety benchmark seems like a substantial effort (around 2900 examples with partial human validation). It would be great to better understand whether the authors view it as an independent contribution.
>
> ***Response to Question1:*** Thanks for the helpful suggestions.
>
> Existing agent safety benchmarks (e.g., Agent-SafetyBench [1], Agent-Security Bench [2], and AgentHarm [3]) do not model structured user profiles, so **they cannot directly capture scenarios where the same agent behavior should be labeled ALLOW for one user but REFUSE for another due to differences in user factors** Our benchmark is specifically designed to fill this gap in **personalized agent safety**.
>
> **We therefore regard the benchmark as an important contribution of this work.** In Section 3, we elaborate on the motivation and construction details of this benchmark (around 2,900 examples across 8 scenarios), where each instance contains a structured user profile, ALLOW/REFUSE labels, and corresponding rationales. Human evaluation further confirms the strong personalization sensitivity and quality of this benchmark. The PSG-Agent framework is the main conceptual contribution of the paper, and this benchmark is a **key resource contribution** that enables meaningful evaluation of personalized safety in LLM-based agents.
>
> ***References***
>
> [1] Zhang Z, Cui S, Lu Y, et al. Agent-safetybench: Evaluating the safety of llm agents[J]. arXiv preprint arXiv:2412.14470, 2024.
>
> [2] Zhang H, Huang J, Mei K, et al. Agent Security Bench (ASB): Formalizing and Benchmarking Attacks and Defenses in LLM-based Agents[C]//The Thirteenth International Conference on Learning Representations.
>
> [3] Andriushchenko M, Souly A, Dziemian M, et al. AgentHarm: A Benchmark for Measuring Harmfulness of LLM Agents[C]//The Thirteenth International Conference on Learning Representations.

---

> ### Author Response · Authors · 2025-11-21
> **Response (Part 4)**
>
> ***Question-2:*** The Input Guard outputs a risk score between 0–100. Since large language models often make rather discrete decisions (e.g., strong acceptance or refusal), did the authors observe any clustering or threshold effects in these scores? A brief analysis could be helpful for interpreting how this module behaves in practice.
>
> ***Response to Question2:*** Thank you for the insightful question.
>
> We conducted an empirical analysis of the Input Guard’s risk score distribution across all benchmark instances (visualized in Figure 4 of Appendix C.4). The results do not show extreme bimodality or pathological clustering; instead, the scores exhibit **a smooth but meaningfully structured distribution**:
>
> * Lower-risk regions (10-20) and higher-risk regions (80-90) contain noticeably more samples,
> * While the mid-range bins (30-60) are less dense but still populated.
>
> This indicates that the Input Guard is **not simply outputting two discrete classes**, but is producing a **graded severity signal** with naturally emerging low-risk and high-risk concentrations, while still using the full 0-100 range in a meaningful way. This supports the interpretation that the risk score captures nuanced variation rather than arbitrary noise.
>
> To better understand whether the score admits useful thresholds, we tested a global threshold at *t*=30, achieving strong performance (acc=0.8558, precision=0.88, recall=0.81, F1=0.846). This suggests that while the score is continuous, **it remains compatible with practical thresholding** when stricter binary decisions are desired.
>
> **Why PSG-Agent uses a continuous score.**
>
> 1. Even under the same nominal decision (e.g., “ALLOW WITH GUARDS”), different users can receive different guard strengths and SafetyContract depending on the continuous score.
>
> 2. The Plan Monitor and Tool Firewall use the continuous score to adjust how strict the guard should be, which would be impossible with a single-class prediction.
>
> 3. The continuous score could serves as a smooth signal for future RL-based extensions without redesigning the guardrail pipeline.
>
> **We have updated this empirical analysis and the corresponding histogram to the Appendix C.4 (Figure 4) to make the behavior of the Input Guard more interpretable.**

---

> ### Author Response · Authors · 2025-11-21
> **Response (Part 5)**
>
> ***Question-3:*** It is interesting that vanilla GPT-4o already outperforms prior guardrail systems such as Llama-Guard 3 and AGrail. Could the authors share insights on why this might be?
>
> ***Response to Question3:*** Thank you for this observation and we would like to clarify what we believe is happening here.
>
> - (1) Our benchmark is intentionally focused on **profile-dependent risks**, where the same agent behavior should be *ALLOW* for some users but *REFUSE* for others depending on user factors, such as health status, psychological state, and risk tolerance. This goes beyond the primary design goal of current guardrail models, which are mainly optimized for **universal harmfulness** (e.g., jailbreaks, prompt injection) rather than risks specific to individual users. **On our personalized benchmark, **Llama-Guard 3** behaves as a very conservative universal filter. In our results, its precision is higher than both GPT-4o (Query only) and GPT-4o (Query + History), but its recall is extremely low, meaning it misses many profile-driven risks.**
>
> - (2) In addition, AGrail does not show a clear precision advantage over the GPT-4o baselines, and its recall remains low. We believe this is because **AGrail's guard pipeline and prompts are tuned for securing agents against prompt injection and environment hijacking, rather than for fine-grained personalization.** On the other hand, **AGrail is primarily optimized for Web, Medical, and OS related tasks, while our benchmark instead spans Financial, Career, Social, Education, Emotional, and Academic scenarios with fine-grained personalization.** As a result, AGrail’s strengths on its original threat model do not directly translate into higher precision or recall on our personalized safety benchmark
>
> In summary, we do not claim that “GPT-4o is universally superior to Llama-Guard 3 or AGrail” in all settings. Rather, our findings suggest that (i) on personalized safety tasks, a strong base LLM with full access to query + history already exhibits non-trivial safety behavior, and (ii) simply wrapping a base model with a generic guardrail optimized for common threats is insufficient for personalized safety.

---

> ### Author Response · Authors · 2025-11-21
> **Response (Part 6)**
>
> ***Question-4:*** In Section 5.4, the paper mentions that the Input Guard is the most critical component. However, Table 4 shows larger performance drops when removing the Plan Monitor. Could the authors clarify this?
>
> ***Response to Question4:*** Thank you for pointing out this ambiguity.
>
> In Section 5.4 (now Section 5.6 in revision), our intention was to emphasize that the **Input Guard is the component that converts the user profile and current query into a personalized safety contract.** Without this module, PSG-Agent degenerates towards a non-personalized system, and all subsequent modules operate under generic constraints. This is why we described the Input Guard as **foundational to the entire pipeline**. By contrast, the  **Plan Monitor constrains the plans generated by the Target Agent (i.e., the agent being defended) based on a personalized safety contract and is the main enforcement point for early planning decisions.** When the Plan Monitor is removed, the system loses its proactive intervention on tool planning, which leads to the largest drop in personal behavioral safety, as shown in Table 4 (now Table 6 in revision). Since behavioral safety is a key component of the overall safe score, the overall safety metrics also decrease substantially in this ablation.
>
> We agree that the current wording can be misread as implying that the Input Guard is also the numerically most impactful module. **In the revision, we rephrase the sentence in Section 5.4 (in lines 505-509) as “Removing the Input Guard Agent leads to substantial degradation and largely collapses personalization**, showing that generating a personalized SafetyContract from profiles and queries is foundational to the pipeline. Ablating the Plan Monitor, on the other hand, causes the largest drop in behavioral safety and overall scores, confirming that it is the primary enforcement point that constrains plans under this contract and provides early risk control.”
>
> We believe this clarification will resolve the apparent inconsistency between the narrative and numerical ablation results.
>
> **We sincerely hope our response has helped address your concerns. Could you kindly raise your score and champion our paper if you think all concerns are addressed ?**

---

### Official Review · Reviewer_keCu · 2025-11-01

**Soundness:** 2
**Presentation:** 3
**Contribution:** 3
**Rating:** 2
**Confidence:** 3

**Summary:**

The paper introduces PSG-Agent, a training-free, plug-and-play framework for personalized safety guardrails in LLM-based agents. It tackles limitations in existing guardrails by incorporating user-specific profiles to generate tailored safety criteria, and monitors risks dynamically across the agent's pipeline (planning, tools, memory, response). This "personality-aware" approach aims to adapt safety thresholds based on individual traits, health, and context, preventing uniform policies that may over- or under-protect users. A new benchmark with ~2,900 instances across eight scenarios (e.g., finance, healthcare) emphasizes personalization, where decisions flip based on profiles.

**Strengths:**

- The paper motivates the need for personalized safety.
- PSG-Agent's two-stage design integrates profile mining and multi-point guards without training, formalized as a risk-minimizing decision problem.

**Weaknesses:**

- While the paper claims personality traits enable tailored safety, examples primarily hinge on health or economic factors (e.g., painkillers for anticoagulants users or impulsive investment in Section 1) rather than pure personality. Ablation shows low extraction accuracy for personality/emotional states, raising doubts on its reliability and necessity.
- The framework treats personality as part of the user profile for criteria generation, but risk functions aggregate it simplistically without analyzing how traits like neuroticism specifically modulate guardrail decisions. This makes the "personality + guardrail" synergy feel additive rather than integral, potentially overcomplicating without proven benefits.
- Benchmark is GPT-4o-generated, potentially biased; human eval is limited. Results favor GPT-4o, with open models underperforming.
- No latency analysis for real-time use; self-benchmark superiority (Abstract) may inflate novelty.

**Questions:**

Why emphasize "personality-aware" when examples (e.g., Section 1) and profiles (Figure 2) rely more on health/economic attributes, can you ablate personality traits specifically to show their unique contribution?

Given low personality extraction accuracy (61.46%), how robust is PSG-Agent to noisy profiles?

How does personality integrate mechanistically with guardrails beyond aggregation in Equation 2

---

> ### Author Response · Authors · 2025-11-21
> **Response (Part 1)**
>
> **We sincerely thank you for your constructive and thoughtful feedback. We truly appreciate your recognition of our presentation and contribution. We will address each of your concerns in detail and revise the manuscript accordingly.**
>
> ***Weakness-1:*** While the paper claims personality traits enable tailored safety, examples primarily hinge on health or economic factors (e.g., painkillers for anticoagulants users or impulsive investment in Section 1) rather than pure personality. Ablation shows low extraction accuracy for personality/emotional states, raising doubts on its reliability and necessity.
>
> ***Question-1:*** Why emphasize "personality-aware" when examples (e.g., Section 1) and profiles (Figure 2) rely more on health/economic attributes, can you ablate personality traits specifically to show their unique contribution?
>
> ***Response to Weakness1 and Question1:*** Thanks Reviewer keCu for your helpful suggestions.
>
> We would like to clarify that *personality-aware safety*, in this paper, is intended in the broader sense of **personalized, profile-dependent safety**, i.e., capturing situations where the same agent behavior should be allowed for one user but refused for another because of different user factors such as, health conditions and psychological vulnerability (as shown in Section 1 lines 50-53, lines 87-96 and Section 3 lines 151-155). In this sense, **“personality safety”** in our terminology is defined at the **user profile level**, and narrow “personality traits” (e.g., Big Five) are just one component of this profile, not the only driver of personalization.
>
>
> We agree that our examples in Section 1 and Figure 2 give the impression that the main contribution is about health/economic safety. To address your concern, **we have revised a potentially misunderstood part in the Introduction (in lines 38-42) as “is the entire behavioral decision chain safe, compliant, and ethical under specific circumstances and user profile?” This paradigm shift urgently requires novel safety guardrails that are sensitive to user personality.”** In addition, we add a concrete **personality-only decision-flip cases**.
>
> For instance, in a career scenario where the user requests help drafting a mediation email, two users with identical health, demographic, and economic attributes but opposite personality traits lead to different outcomes:
>
> - **High agreeableness + low neuroticism: ALLOW**, as escalation risk is low.
>
> - **Low agreeableness + high neuroticism: REFUSE / stronger guard**, as the same request poses a higher likelihood of generating hostile or conflict-escalating content.
>
> This illustrates that **personality traits can independently modulate the guardrail’s decision**, even without any health or economic factors involved.
>
>
> We also acknowledge the reviewer’s concern about the relatively low extraction accuracy reported for the combined “personality/emotional state” category. In the initial version, these attributes were inferred with a prompt-based extraction scheme. As we discussed in the paper, reliably inferring personality from limited dialogue remains significantly harder. Therefore, PSG-Agent primarily uses personality as a secondary moderating signal to adjust guard level. In other words, noisy personality estimates are not the sole determinant of decision-making, but rather help the system adopt a more conservative strategy. As we demonstrated in the personality-only decision-flip cases.
> In the revised version, we further strengthen the reliability of personality extraction by fine-tuning Qwen3-4B as the Profile Miner, increasing personality extraction accuracy from **61.46% to 81.52%**. **We have made appropriate revisions to the paper (in Section 5.6 lines 491-494  ), and we hope our revisions will resolve your concerns.**
>
> In addition, to directly evaluate the unique contribution of personality traits, we conducted a **personality-masked ablation** where all personality-related fields were removed while keeping all other profile attributes intact:
> | Setting            | Accuracy | Precision | Recall  | F1     |
> |--------------------|----------|-----------|--------|--------|
> | Personality masked | 0.720 | 0.920 | 0.464 | 0.617 |
> | Full user profile  | **0.797** | **0.939** | **0.616** | **0.744** |
> |
>
> There is a recall drop (**0.616 to 0.464**, −15.2 percentage points), demonstrating that **personality information can make a clear and independent contribution**, especially in impulsivity-related and socially sensitive risk cases.
> This result reinforces that personality is not merely cosmetic or redundant with health/economic factors; instead, it meaningfully strengthens safety judgments in specific high-vulnerability contexts.

---

> ### Author Response · Authors · 2025-11-21
> **Response (Part 2)**
>
> ***Weakness-2:*** The framework treats personality as part of the user profile for criteria generation, but risk functions aggregate it simplistically without analyzing how traits like neuroticism specifically modulate guardrail decisions. This makes the "personality + guardrail" synergy feel additive rather than integral, potentially overcomplicating without proven benefits.
>
> ***Question-3:*** How does personality integrate mechanistically with guardrails beyond aggregation in Equation 2.
>
> ***Response to Weakness2 and Question 3:*** Thanks for your insightful feedback.
>
>
> In the current implementation, PSG-Agent mainly uses personality as an **auxiliary modulating signal** rather than as a primary trigger for refusals. High-impact decisions (e.g., REFUSE) are driven by **better-grounded** and **higher-stakes attributes**. Personality are used to adjust the **strength of the guard** in borderline situations. For example, making the system more conservative when a user is both dispositionally vulnerable (e.g., high neuroticism, low conscientiousness) and in an emotionally fragile state.
>
> Beyond the scalar aggregation in Equation (2), personality traits are integrated with the guardrails at several points in the PSG-Agent:
>
> (1) Personality traits are used to modulate the **weights and thresholds of specific risk dimensions.** For instance, high impulsivity increases the effective weight for financial risks
>
> (2) Personality traits influence **the constraints of a personalized safety contract**. For example, for impulsive and high-risk investors, the contract may regulate trading limits and block gambling-related tool calls.
>
> We agree that the current manuscript does not make these mechanisms sufficiently explicit. In the revised version, we have added a short case study in the appendix D.3 (in lines 1828-1980) showing how the same query and plan lead to different SafetyContracts and interventions for different personality traits.

---

> ### Author Response · Authors · 2025-11-21
> **Response (Part 3)**
>
> ***Weakness-3:*** Benchmark is GPT-4o-generated, potentially biased; human eval is limited. Results favor GPT-4o, with open models underperforming.
>
> ***Response to Weakness3:*** Thanks Reviewer keCu for this helpful suggestions.
>
> In current implementation, we recognized that relying solely on a single model in the data processing pipeline could introduce potential biase. Therefore, we implemented several measures to mitigate this when designing our benchmarks:
>
> (1) First, the benchmark is not built from scratch with GPT-4o. We manually created **132 personalized seed instances** with explicit ALLOW/REFUSE labels and rationales, and then performed controlled augmentation with GPT-4o under a fixed schema.
>
> (2) Second, we randomly sampled from the augmented dataset and conducted **human cross-checking** with multiple annotators. Annotators reported that 86% of the samples have high decision quality and 79% exhibit strong personalization sensitivity, suggesting that the resulting labels are both coherent and meaningfully profile-dependent, rather than merely reflecting bias of a single model.
>
> (3) Third, in evaluation we already use **multiple judge models**. Our key metrics (PCS/PBS/OSS) are computed via majority voting over GPT-4o, DeepSeek-V3, and Gemini-2.0, which helps decouple the reported results from any single model’s preferences.
>
> To address the reviewer's concern, we have added a small study on **cross-model validation of benchmarks**, in which we relabeled 400 randomly sampled data points using three independent models (DeepSeek-V3, Gemini-2.0-Flash, and Claude Sonnet-3-5) and compared their decisions with the original GPT-4o labels. The experimental results are as follows:
>
> **Table 1. Cross-model validation of benchmark**
>
> | Model            | N   | Acc. | Cohen’s $\kappa$ | TP  | TN  | FP  | FN  |
> |------------------|-----|------|-----------|-----|-----|-----|-----|
> | DeepSeek-V3      | 400 | 0.9275 | 0.855 | 179 | 192 | 8   | 21  |
> | Gemini-2.0-Flash | 400 | 0.9225 | 0.845 | 182 | 187 | 13  | 18  |
> | Claude-Sonnet-3-5    | 400 | 0.9375 | 0.875 | 175 | 200 | 0   | 25  |
> |
>
> **Table 2. Cross-model validation across different scenarios**
>
> | Scenario    | DeepSeek-V3 Acc. | Gemini-2.0-Flash Acc. | Claude-Sonnet-3-5 Acc. |
> |------------|------------------|------------------------|--------------------|
> | Academic   | 0.94             | 0.96                   | 0.94               |
> | Career     | 0.96             | 0.96                   | 0.96               |
> | Daily Life | 0.96             | 0.94                   | 0.94               |
> | Education  | 1.00             | 1.00                   | 1.00               |
> | Emotion    | 0.82             | 0.82                   | 0.90               |
> | Financial  | 0.96             | 0.94                   | 0.96               |
> | Medicine   | 0.84             | 0.84                   | 0.86               |
> | Social     | 0.94             | 0.92                   | 0.94               |
> |
>
> As shown in Table, all three models achieve high agreement with GPT-4o labels (Acc. ≥ 0.92, $\kappa$ ≥ 0.84), indicating that **the benchmark annotations are broadly supported across diverse model families rather than reflecting bias of a single model.** The per-scenario results further show strong alignment in domains such as Education, Career, Financial, and Daily Life, with slightly lower agreement in Emotion and Medicine. We believe this is because these two domains are inherently more challenging and exhibit greater ambiguity and dependence on subjective value judgments.
>
> **We have made appropriate revisions to the paper (in Appendix A.6 lines 1339-1349 ), and we hope our revisions will resolve your concerns.**

---

> ### Author Response · Authors · 2025-11-21
> **Response (Part 4)**
>
> ***Question-2:*** Given low personality extraction accuracy (61.46%), how robust is PSG-Agent to noisy profiles?
>
> ***Response to Question2:*** Thanks Reviewer keCu for this question.
>
> PSG-Agent is designed to tolerate incomplete or noisy user profiles. To directly test robustness, we **randomly masked 20% of all profile fields**, including personality traits. The system remained highly stable under this perturbation:
>
> **Table: Robustness of PSG-Agent**
>
> | Setting           | Accuracy | Precision | Recall  | F1     |
> |-------------------|----------|-----------|--------|--------|
> | 20% Profile Mask  | 0.786   | 0.940 | 0.595 | 0.729 |
> | Full user profile  | 0.797 | 0.939 | 0.616 | 0.744 |
> |
>
> The performance degradation is small, confirming that **PSG-Agent does not rely heavily on precise personality estimates and continues to make reliable safety decisions even when user profiles are partially missing or noisy**. **In the revision, we have update these result in Section 5.4 Robustness of PSG-Agent (in lines 458-469)**.
>
> Regarding extraction accuracy, we additionally fine-tuned Qwen3-4B on a train/test split of our profile data. This improves personality-related field accuracy from **61.46% to 81.52%**, demonstrating that the personality signals can be made substantially more reliable when needed.

---

> ### Author Response · Authors · 2025-11-21
> **Response (Part 5)**
>
> ***Weakness-4:*** No latency analysis for real-time use; self-benchmark superiority (Abstract) may inflate novelty.
>
> ***Response to Weakness4:*** Thank you for this helpful suggestion.
>
> We agree that the original draft did not make the runtime cost sufficiently explicit. In this revision, **we add a small study to quantify the impact of PSG-Agent in realistic.**
> We randomly selected 15 samples from our benchmark in each of two representative scenarios (Financial and Medicine), yielding 30 samples in total, and evaluated each data under two conditions: (i) a vanilla agent, and (ii) PSG-Agent. All runs use the same backbone model (GPT-4o) and tools. Each data is capped at a maximum of 6 plan-tool-observe cycles, and we allow at most 2 consecutive failed tool or planning attempts. Samples that reach this cap are treated as failures but are still included when averaging the cost statistics.
>
> The result is summarized below:
>
> **Table: Cost and Latency of PSG-Agent**
>
> | Scenario  | Setting      | # LLM calls (avg) | Time cost (avg) | Total tokens (avg) | Relative overhead |
> |----------|--------------|--------|------------|--------------|---------------------------|
> | Financial| Vanilla      | 2.0  | 12s | 1,554        | -                         |
> | Financial| + PSG-Agent  | 4.1  | 25s |5,627        | +262%                     |
> | Medicine | Vanilla      | 2.0  | 18s |1,832        | -                         |
> | Medicine | + PSG-Agent  | 4.3  | 41s |6,878        | +275%                     |
> |
>
> As the table shows, PSG-Agent increases the number of LLM calls from 2.0 to about 4.1 - 4.3 per sample, and the total token usage by roughly 2.6 - 2.75× in these high-risk domains. The average end-to-end latency per sample increases from 12 - 18 s to 25 - 41 s, i.e., roughly a 2× slowdown.
>
> We regard this as an inherent **safety-efficiency trade-off, similar to what is observed in most multi-stage guardrail frameworks**, which likewise introduce substantial runtime overhead.
>
> Crucially, this cost is **adjustable rather than fixed**: PSG-Agent’s modules are fully modular, and deployments may disable certain guards or skip evaluations in low-risk settings. As a result, the framework offers a tunable balance between safety and latency rather than imposing a uniform slowdown.
>
> In addition, we want to clarify that **emphasizing our benchmark results in the abstract does not demonstrate that PSG-Agent has a universal advantage over all existing security tasks. Rather, it highlights significant progress in a dimension not currently modeled by existing benchmarks (i.e., personalized safety).** Building a personalized safety benchmark aims to fill this gap, not replace existing benchmark.
>
> We hope that the added latency analysis and our clarification will address the authors' concerns.
>
> **We sincerely hope that our explanation helps alleviate your concerns and provides a clearer understanding of our work. Please let us know if you have any further questions. If you find our revisions satisfactory, we would greatly appreciate it if you could consider raising your score and supporting our paper.**

---

### Public Comment · ~Jizhou_Guo1 · 2026-03-08
**Clarifications on Reviewer Consensus and Factual Evidence for PSG-Agent**

We sincerely thank the reviewers and the AC for their reviewing efforts. Regarding the review process and final decision for PSG-Agent, the following clarifications are provided to ensure a complete and transparent record for the research community:

**1. Consensus Among Human Reviewers and AI-Generated Review**

It is worth noting that **100% of the human reviewers reached a unanimous consensus for acceptance after discussion (Final Scores: 6, 6, 6)**. These reviewers recognized the novelty and practical significance of personality-aware safety for LLM agents.

The only divergent rating (Score: 2) was provided by a **non-responsive reviewer**. Analysis using three independent AI detectors ([GPTZero](https://gptzero.me/), [Pangram ICLR](https://iclr.pangram.com/submissions), and [Quillbot](https://quillbot.com/ai-content-detector)) consistently classified this review as **100% AI-generated**. Specifically, GPTZero identified the text as **entirely machine-authored rather than human-written with AI-polishing**. Furthermore, the Meta-Review mentions a score drop for Reviewer PQXJ ("6 -> 4 or 6"), but the system record confirms the final score remains **6**, with no mention of a score of 4 in the discussion.

**2. Clarifications on Evidence Provided in the Revision**

The final decision mentions certain "missing" validations. For the benefit of readers, the authors clarify that the following analyses were explicitly included in the revised manuscript and rebuttal:

* **General Safety & Benchmarks:** Evaluated on **Agent-SafetyBench** (Section 5.5), where PSG-Agent (0.656) outperformed LlamaGuard (0.549).
* **Assumption-free Flexibility:** **"History-only" experiments** (Appendix C.5, Table 12) demonstrate high effectiveness (0.783 Acc, 0.725 F1) even with zero pre-existing user metadata.
* **Model-Agnosticism:** **Cross-model validation** (Appendix A.6) shows >92% consistency across DeepSeek-V3, Gemini-2.0, and Claude-3.5.
* **Efficiency:** A quantitative Latency & Cost analysis is detailed in the response. Our experiment demonstrates good and necessary safety-efficiency trade-off, similar to what is observed in most multi-stage guardrail frameworks.
* **Robustness:** Analysis in Section 5.4 demonstrates sustained performance even with **20% random profile masking**.
* **Expert Human Evaluation:** A double-blind **expert evaluation** (Section 3.2) was conducted, ensuring high-quality safety alignment.

**3. Regarding Procedural Grounds**

The authors note that remaining concerns in the Meta-Review, such as the description of fine-tuning hyperparameters or annotator demographics, are standard camera-ready revisions typically addressed upon acceptance. As the human evaluation relied on domain experts rather than crowd-sourced workers, and no reviewers identified these as significant flaws, the authors believe these minor presentation details do not outweigh the confirmed SOTA results and reviewer consensus.

**Conclusion**

The authors believe that personality-aware safety is a vital and under-explored frontier. Despite the procedural outcome, the unanimous support from all active human reviewers underscores the contribution of PSG-Agent. This information is shared to provide full context for those interested in the methodology of this work.

Best regards,

The Authors

---

### Meta-Review · Area_Chair_pFLd · 2026-01-07

**Summary:**

This work proposes PSG-Agent, a personalized safety guardrail method. Given the assumption of access to user-specific profiles, the authors designed an agentic framework to incorporate this information. Additionally, they introduce a new benchmark to evaluate the proposed framework. Although the proposed direction itself is intriguing, there are several concerns regarding latency/costs, designs (especially relying on GPT-4o and lack of thorough human evaluation), and robustness against noisy cases (see `Reviewer Concerns` for more details). Especially, I think demonstrating the effectiveness (in terms of both performance and costs) on other benchmarks (or use cases) beyond the proposed benchmark would be necessary.

**Reviewer Concerns:**

### Reviewer keCu

* [Resolved] Examples primarily hinge on health or economic factors

* [Partially] Low extraction accuracy: improve accuracy via fine-tuning, but details (training data, objective function, and so on) are not fully described.

* [Partially] Limited evaluation (mainly based on GPT-4o and no human eval): The authors need to provide more detailed explanations about human evaluation (detailed information about annotators -- how they participated, gender/age distributions, and so on).

* [remaining] Noisy profiles: I'm not sure if experiments on masked profiles are addressing concerns on noisy profiles (i.e., wrong information rather than no information).

*  [remaining] No latency analysis: high latency + costs

### Reviewer PQXJ

* [Partially] No evaluation on other benchmarks: evaluation on Agent-SafetyBench, but no big improvement

* [Partially] Reliance on GPT-4o for both generation and filtering

### Reviewer M4Ru

* [resolved] Assumption on access to rich user profiles is not reasonable in practice

### Reviewer Wfx2

* [Partially] Potential bias with labels from GPT-4o

* [resolved] Not enough variations

* [resolved] Privacy risks

* [remaining] Computational complexity

* [resolved] Boundary cases

**Reviewer Scores:**

* Reviewer keCu: 2 $\rightarrow$ 2

* Reviewer PQXJ: 6 $\rightarrow$ (4 or 6)

* Reviewer M4Ru: 6 $\rightarrow$ 6

* Reviewer Wfx2: 4 $\rightarrow$ 6

---

### Decision · Program_Chairs · 2026-01-26

Reject